# Corneal Refractive Surgery Considerations in Patients on Dupilumab

**DOI:** 10.3390/jcm11123273

**Published:** 2022-06-08

**Authors:** Majid Moshirfar, Tanner Seitz, Brianna Ply, Yasmyne C. Ronquillo, Phillip C. Hoopes

**Affiliations:** 1Hoopes Vision Research Center, Hoopes Vision, 11820 S. State St., Ste. 200, Draper, UT 84020, USA; bply@hoopesvision.com (B.P.); pch@hoopesvision.com (P.C.H.); 2John A. Moran Eye Center, University of Utah School of Medicine, Salt Lake City, UT 84132, USA; 3Utah Lions Eye Bank, Murray, UT 84107, USA; 4Midwestern University, Glendale, AZ 85308, USA; tseitz98@midwestern.edu

**Keywords:** dupilumab, Dupixent, monoclonal antibody, LASIK, PRK, SMILE, cornea, meibomian gland, goblet cells, dry eye, atopy

## Abstract

Dupilumab is a biologic approved by the United States Food and Drug Administration (US FDA) for the treatment of atopic dermatitis. While it is an effective medication for eczema, ocular side effects are common in patients receiving dupilumab therapy. Greater consideration is needed when evaluating these individuals for corneal refractive surgery. Dupilumab patients may suffer from atopy, a condition that also merits consideration in those desiring refractive surgery. Additional testing and careful consideration are needed, as these patients have an increased risk of dry eye syndrome, keratoconus, cataracts, diffuse lamellar keratitis, viral keratitis, and perioperative infection. This commentary discusses the current understanding of dupilumab ocular side effects and investigates factors to consider when evaluating these patients for corneal refractive surgery.

Dupilumab (Dupixent) is a monoclonal antibody that was approved in 2017 by the US FDA for the treatment of moderate to severe atopic dermatitis and asthma maintenance therapy. Dupilumab blocks interleukins 4 and 13, preventing the differentiation of undifferentiated T-cells into allergy-regulating type 2 helper T-cells [1,2]. Notable improvement in atopic dermatitis symptom severity has been documented since dupilumab entered the market.

US FDA reports of 529 patients show that ocular side effects at 16 weeks of dupilumab treatment include conjunctivitis (10%), blepharitis (<1%), dry eye (<1%), eye pruritis (1%), and keratitis (<1%), with an average time of symptom onset being 6 ± 5.5 weeks [3,4,5]. Due to these side effects, the dupilumab insert suggests discussing all medical conditions, including eye problems, with a healthcare provider.

Interleukins 4 and 13 have been implicated in dry eye symptoms in dupilumab patients. Interleukin-4 is integral to the production of meibum, and blocking its function causes meibomian gland dysfunction (MGD). Upwards of 25% of patients receiving dupilumab suffer from this condition, and symptoms appear after six months of dupilumab use [6,7]. IL-13 has been shown to stimulate goblet cells, and inhibiting its function can lead to hypoplasia or complete absence of these cells in dupilumab patients [8,9,10]. Goblet cell dysfunction causes decreased mucin on the ocular surface. Mucin maintains the wettability of the ocular surface and contributes to tear film stability [11]. Patients on dupilumab may experience dry eye symptoms due to mucin deficiency, disrupting the tear film layer.

Patients on dupilumab desiring corneal refractive surgery (CRS) undergo a litany of tests, in addition to a complete history, comprehensive eye examination, and screening for glaucoma, corneal diseases, and other pre-existing eye or systemic diseases. These tests include corneal staining, which may reveal the characteristic MGD pattern of punctate epitheliopathy and erosion. Eye care providers should be increasingly vigilant to the information disclosed by meibography, corneal staining, tear break-up test, conjunctival impression cytology, and dry eye questionnaires because of increased susceptibility in dupilumab patients to tear problems, MGD, and conjunctivitis (Table 1).

In addition, it is essential to be cognizant of the underlying medical issue in patients on dupilumab. They may suffer from atopy, a genetic tendency to develop diseases such as asthma, atopic dermatitis, and allergic rhinitis. Thorough screening for keratoconus, ectasia, and subtle posterior subcapsular cataract is necessary as atopic patients have a higher predisposition for these conditions. Eye care professionals need to perform a thorough slit lamp examination in conjunction with detailed tomography and topography screenings during CRS evaluation of this patient population. Assessment of eye rubbing habits should also be included in the evaluation, as the risk of keratoconus is increased with chronic eye rubbing [12] (Table 1).

Dupilumab patients also have higher staphylococcal bacterial counts on the skin, lid margin colonization, and increased risk of viral keratitis and diffuse lamellar keratitis if undergoing LASIK surgery [13]. Bacterial colonization of the eyelid margin increases the risk of perioperative corneal infection without proper sanitation practices [14]. The patient’s informed consent should include the additional risks and post-operative complications after CRS such as worsening dry eye.

It is important for eye care physicians and referring providers to consult with the managing dermatologist or allergist concerning the stability of the disease prior to elective procedures such as corneal refractive surgery. A plan regarding the maintenance of dupilumab and continuity of care should be in place, including artificial tears, punctal plugs, and topical administration of cyclosporine and lifitegrast as needed [15]. Most patients do not need to discontinue dupilumab while adverse effects are managed. Only severe cases or cases refractive to post-refractive surgery treatment may require discontinuation of dupilumab [16].

## Conclusions

Dupilumab was approved by the US FDA for the treatment of atopic dermatitis and has been effective in disease management. It is not uncommon for these patients to suffer from other atopic diseases. Atopy and dupilumab place patients at increased risk for cataracts, keratoconus, dry eye, viral conjunctivitis, and postoperative complications. These patients may seek corneal refractive surgery, presenting complex cases for eye care professionals. While the evidence is limited, there is no absolute contraindication to CRS, even if a patient is on dupilumab treatment. However, these cases require an in-depth evaluation, including greater attention to meibography, topography, corneal staining, goblet cell function, and awareness of increased perioperative infection.

## Figures and Tables

**Table 1 jcm-11-03273-t001:** Clinical Findings, Eye Examinations and Tests for Evaluation of Atopic Patients on Dupilumab Considering Corneal Refractive Surgery.

Clinical Finding	Examinations and Tests
Conjunctivitis	Dupilumab-associated conjunctivitis necessitates special conjunctival staining and careful examination of bulbar and palpebral conjunctival surfaces
Meibomian Gland Dysfunction (MGD)	Meibography, tear film break-up time, Schirmer test
Goblet Cell Dysfunction	Conjunctival Impression Cytology
Viral keratitis	Slit lamp examination, corneal staining Atopic patients are at higher risk of developing viral infections such as herpetic keratitis when on dupilumab
Staph marginal keratitis	Slit lamp examination, Gram stain, culture, and sensitivity More predisposed to blepharitis, can develop marginal keratitis during the perioperative stage of CRS with a high risk of corneal infection
Cataract	Slit lamp examination, specifically for posterior subcapsular cataract in atopic patients
Keratoconus	Corneal tomography and topography atopic patients are more predisposed to corneal ectasia
Diffuse Lamellar Keratitis (DLK)	Slit lamp examination Atopic patients have a greater predisposition to DLK after LASIK surgery

## Data Availability

Data sharing is not applicable to this article as no datasets were generated or analyzed during the current study.

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
