# Peer review of "Corneal Refractive Surgery Considerations in Patients on Dupilumab"

_jcm, 2022, doi:10.3390/jcm11123273_

Round 1

Reviewer 1 Report

The authors presented an interesting commentary on Dupilumab and corneal refractive surgery. The findings are worth reporting, but the authors should improve the manuscript for English language and punctuation and revise the manuscript for use of abbreviations (the i.e. line 10 and 22 FDA should be explained as “Food and Drug Administration”).

Author Response

Thank you for the important suggestions. We have added United States Food and Drug Administration (US FDA) in line 10. This would also make the acronym US FDA in line 23 clear. 

We have gone over the manuscript for English editing and submitted the revised version.  

Reviewer 2 Report

The authors revealed and discussed the ocular side effects in patients receiving dupilumab therapy. They recommend some routine eye medical examinations in dupilumab patients, or especially in corneal refractive surgery consideration. Author also discussed the underlying mechanism that affected and implicated in dry eye symptoms. The manuscript is well-written according to the topic. The reference citations and the discussion were well. I have no concerns.

Author Response

Thank you for your time and expertise in reviewing this article. This article will be useful to clinicians, referring health care providers and patients. 

Reviewer 3 Report

The paper entitled “Corneal Refractive Surgery Considerations in Patients on Dupilumab” is a viewpoint based on the ocular effects of this therapy, which is used in the treatment of atopic dermatitis. The report states that greater consideration is needed when evaluating these individuals for corneal refractive surgery as these patients have an increased risk of dry eye syndrome, keratoconus, cataracts, diffuse lamellar keratitis, viral keratitis, and perioperative infection. The paper is interesting and of potential clinical interest.

Specialists can sometimes make clinical decisions without assessing all aspects of the patients. Although the advantages of no-spectacle use after surgery can be important to some, certain patients can experience negative side effects that could have been avoided without surgery. Patients must be properly informed of all possible effects after refractive surgery, especially those patients at risk. As mentioned by the authors, chronic dry eye, corneal pathologies, and other ocular surface conditions can worsen after refractive surgery. This section could be enhanced with greater information and additional relevant citations.

With regards to patients that preferably should not undergo refractive surgery, the authors should include patients with ocular hypertension, glaucoma, or at risk of glaucoma due to the difficulties in assessing true IOP in iatrogenic thinner corneas.

The study has been correctly planned. It is well written and of clinical interest. The study provides valuable suggestions and adds to current literature in this field.

Author Response

Thank you for your comments and suggestions.

  1. With regards glaucoma, ocular hypertension, risk of glaucoma and thin corneas, these diseases and factors are included in the usual screening for refractive surgery, whether or not the patient is on dupilumab therapy.  We have added lines 44-45 to refer to the usual screening evaluation prior to refractive surgery "... in addition to complete history, comprehensive eye examination and screening for glaucoma, corneal diseases and other pre-existing eye or systemic diseases.
  2. The patient's informed consent prior to refractive surgery includes a discussion on the complications that the reviewer mentioned-- worsening dry eye, ocular surface disease. Thus, we did not delve into this aspect in detail, as most articles on refractive procedures discuss the standard risk/benefits of LASIK surgery and its complications. We have added lines 67-68 "The patient's informed consent should include the additional risks and postop complications after CRS such as worsening dry eye".
  3. We focus on the effects of dupilumab on patients who need more detailed pre-op evaluations and discussion of the added risks for the patient before mutually deciding on refractive surgery.  Post-op evaluations would also entail more details as noted in the article. Discussions with the managing physician is also important, as pointed out in the discussion.